# Delving into L2 Learners' Perspective: Exploring the Role of Individual Differences in Self-Evaluation of L2 Speech Learning

Yui Suzukida [1,2]

1    School of Medicine, Juntendo University, Tokyo 113-8421, Japan; y.suzukida.fz@juntendo.ac.jp
2    Institute of Education, University College London, London WC1E 6BT, UK

**Abstract:** Misalignment between second language (L2) self-perception and actual ability is often observed among L2 learners. In order to further understand this phenomenon, the current study investigated how the roles of individual differences (IDs; especially experiential and cognitive IDs) influence the learners' self-assessment accuracy. To this end, L2 speech samples elicited from 97 Japanese learners of English were analyzed via self-evaluation and expert evaluations. Subsequently, learners' IDs profiles, including working memory, phonological memory, implicit learning and auditory processing, were linked to (a) the gap between self- and expert evaluation scores and (b) the type of inaccurate self-evaluation (i.e., overconfident vs. underconfident evaluations). The study illustrates the complex relationships between L2 learners' linguistic knowledge, cognitive abilities, experiential profiles and self-perception.

**Keywords:** self-evaluation; L2 speech; individual differences

## 1. Introduction

Self-assessment in second language (L2) learning has increasingly attracted attention due to its instructional and evaluative potential (Butler and Lee 2010). The process of self-assessment does not simply entail gauging one's own learning progress, but it also fosters autonomy and self-regulatory learning (Dann 2002; Oscarson 1989, 1997; Paris and Paris 2001). Furthermore, it serves as an invaluable tool for increasing learners' awareness of their learning trajectories and performances (Boud 1995; Isbell 2021; Panadero et al. 2017; Schmidt 1990). Despite the benefits of self-assessment for language acquisition, numerous studies have reported that the participants' estimations are often inaccurate (Foote 2010; Suzuki 2015), while others have indicated that learners can accurately estimate their L2 proficiency (Brantmeier and Vanderplank 2008; Luoma 2012; Préfontaine 2013). Inaccuracy in L2 learners' self-perceived judgment of various linguistic aspects suggests that bridging the gap between perceived and actual L2 competence poses a significant challenge. Mismatched perceptions can affect learners' confidence, which in turn impacts on their willingness to communicate (de Saint-Léger 2009; de Saint-Léger and Storch 2009). Such discrepancies in self-evaluations can lead to missed opportunities for skill enhancement in and outside of the classroom.

The degree of alignment between self-evaluation and objective measures appears to vary across language skills. In their meta-analysis, Li and Zhang (2021) revealed that receptive skills, such as listening and reading, generally had a stronger correlation with actual performance in comparison to productive skills such as writing and speaking. However, these findings hint at underlying individual differences (IDs) factors that could influence the accuracy of self-assessments. Delving deeper into these influential factors, Butler and Lee (2010) posited that the validity of self-assessments could be influenced by the specific domain being evaluated, the formulation and delivery of assessment items, and crucially, students' individual characteristics.

IDs are of particular interest as they can significantly influence the accuracy of self-assessments. Factors such as proficiency, learning experience, age, memory recall related

to the assessed skills, and learners' first languages (L1s) have been found to play a role (Blanche and Merino 1989; Davidson and Henning 1985; Heilenman 1990; Janssen-van Dieten 1989; Patri 2002; Stefani 1998; Butler and Lee 2006; Ross 1998). Further complicating the issue are psychological and affective factors. For example, learners' anxiety levels associated with language learning can cloud their judgment (MacIntyre et al. 1997). Similarly, self-esteem and motivation can either promote or hinder accurate self-assessments (e.g., AlFallay 2004; Anderson 1982; Dörnyei 2001).

In light of these influences, ensuring the accuracy of self-evaluation in L2 learning becomes both a challenge and a necessity. Recognizing and understanding the variety of factors that might impact on this process is vital. Therefore, the aim of the current study is to explore the IDs factors by focusing on L2 speaking to provide a comprehensive understanding of self-assessment in L2 learning. Through this exploration, language instructors and L2 learners alike can better harness the power of self-assessment, optimizing both the learning process and the outcomes.

## 2. Background

### 2.1. Self-Evaluation in L2: Match or Mismatch?

Insights from psychology studies, such as the work of Burson et al. (2006), have increased our understanding of biases in self-evaluation by demonstrating that individuals, regardless of their skill level, might not accurately grasp how good their abilities are compared to others. An interesting pattern has been observed in these studies: More difficult tasks often lead to a negative bias, while those who are performing poorly may still believe that their assessments are quite accurate. Furthermore, the phenomenon known as the Dunning–Kruger effect, first identified by Kruger and Dunning (1999), adds depth to these findings. This effect is a cognitive bias whereby individuals with limited knowledge or competence in a certain domain overestimate their abilities due to a lack of self-awareness regarding their skill level. Kruger and Dunning proposed that people could not appraise their own competence objectively if they lacked the necessary metacognitive skills. By contrast, those with substantial skills might underestimate their competence due to mistakenly assuming that what is easy for them is easy for everyone else. Dunning et al. (2003) further illustrated this bias, and showed that self-assessments were often not aligned with actual performances, particularly when compared to peers' performances. This misalignment emphasizes the complex nature of self-evaluation and the tendency for individuals to both under- and overestimate their abilities.

In the context of L2 learning, L2 learners naturally develop an awareness of their proficiency and speech patterns guided by individualized goals. For over 30 years, L2 researchers have been interested in understanding L2 learners' awareness and perception of their L2 proficiency (e.g., Heilenman 1990; Isbell and Lee 2022; Janssen-van Dieten 1989) due to its significant value in language acquisition (e.g., Schmidt 1990 for Noticing Hypothesis; Jessner 2008 for Metalinguistic awareness) and potential consequences for learning outcomes (e.g., Butler and Lee 2010; de Saint-Léger 2009 for low willingness to communicate in an L2). In the abundant literature on self-assessment, researchers have identified various parameters that can impact on the accuracy of self-assessments (e.g., Butler and Lee 2010; Gaffney 2018). These include (a) the manner in which questions and assessment items are constructed and conveyed (e.g., Likert scales or descriptive criterion-referenced measures, conducted via L1 or L2), (b) the domain or skill being evaluated (speaking, listening, reading, or writing), and (c) the unique characteristics inherent to L2 learners themselves (i.e., IDs, which are the focus of the current study). With regard to the first factor, the accuracy of self-assessments appears to be greater when the questions are directly aligned with the students' immediate objectives (Butler and Lee 2006); another possibility is that learners exhibit more precision when they assess skills in their first language compared to the target language (Oscarson 1997). In the context of studies of L2 self-assessments, while variations can be observed depending on the skills (e.g., Suzuki (2015) for Can-Do statements for the self-evaluation of reading skill in Japanese; Butler and Lee (2010) for sum-

mative and unit-based English performance; Gaffney (2018) for L2 vocabulary), L2 speech self-evaluation studies have consistently utilized 9- or 10-point Likert scales (Isbell and Lee 2022; Ortega et al. 2022; Trofimovich et al. 2016) or 10-point scales (Saito et al. 2020b), with the exception of "YES/NO" responses to the correctness of word pronunciation in a study by Dlaska and Krekeler (2008). Therefore, the source of discrepancies (or even the differences in the degree of discrepancies in the existing L2 speech self-evaluations) may be due to the type/nature of the linguistic aspect being assessed, and IDs.

*2.2. Assessing L2 Speech*

L2 learners' self-evaluations have been explored across various linguistic domains to date, including listening (e.g., Brantmeier et al. 2012), reading (e.g., Brantmeier 2006; Suzuki 2015), writing (e.g., Yaghoubi-Notash 2012), vocabulary (e.g., Gaffney 2018), and speaking skills (e.g., de Saint-Léger 2009; Dlaska and Krekeler 2008 for pronunciation). Ross (1998) highlighted that the ambiguity or fuzziness of the domain of evaluation may contribute to misestimations. In his meta-analysis, the adult learners' self-assessments of receptive skills, such as listening and reading, tended to be more accurate compared to those for productive skills such as speaking and writing. A recent meta-analysis conducted by Li and Zhang (2021) also indicated a relatively large discrepancy in speech evaluation, suggesting that speaking may be one of the skills that suffers most from incorrect self-evaluations (see Gaffney 2018 for the discussion of ambiguity in terms of vocabulary). In fact, while correlations between learners' self-evaluations and external criteria (such as experts evaluations) generally appear to be moderate ($r = 0.40$–$0.60$) across various aspects of L2 domains, the correlation coefficients of speaking show low or no correlation between self-assessments and objective assessments (e.g., $r = 0.19$ in Ortega et al. (2022); $r = 0.08$ in Trofimovich et al. (2016) for Accentedness; $r = 0.07$ in Saito et al. (2020b); $r = 0.18$ in Trofimovich et al. (2016) for comprehensibility), indicating that L2 learners' understanding of scales such as accentedness and comprehensibility may be somewhat arbitrary (however, see $r = 0.54$ in Isbell and Lee (2022);[1] $r = 0.40$ in Ortega et al. (2022) for instances of the accurate calibration of comprehensibility).

Such miscalibrations can be caused by several factors. Research on L2 learners' metacognition has shown that people's attitudes towards successful verbal communication appear to be skewed toward segmental accuracy (e.g., Derwing and Rossiter 2002; Zoss 2015). For example, Derwing and Rossiter (2002) asked 100 low to intermediate learners of English as a second language (ESL) to identify issues that could cause communication breakdowns. In this case, 40% of the participants were unable to locate the areas of difficulty when communicating in the L2. The rest of the participants (60%) answered that the errors were segmental, with almost no attention being paid to suprasegmental features (e.g., rhythm, intonation, and lexical stress; also see Strachan et al. (2019) for a discussion of learner awareness). Therefore, existing evidence suggests that the types and quality of L2 learners' awareness of the global dimensions of pronunciation may have an impact on the degree of mismatch between self-assessment and objective assessments.

Of note, the Dunning–Kruger effect (Dunning et al. 2003), which has often been observed in psychology studies, has also been observed in the literature on L2 speech (e.g., Patri 2002; Trofimovich et al. 2016; Saito et al. 2020b). For example, a number of studies, including the work of Trofimovich et al. (2016), Li (2018), Saito et al. (2020b), and Isbell and Lee (2022), found that L2 learners with better speaking skills tended to underestimate their comprehensibility and accentedness. Lee and Chang (2005) identified similar patterns in oral presentation abilities.

*2.3. Individual Differences and L2 Speech*

The third factor that is considered to play a role in the miscalibrations is IDs. Research on L2 self-evaluation has investigated several ID factors, such as gender (Pallier 2003), proficiency (e.g., Brantmeier et al. 2012; Strong-Krause 2000), L2 learning experience (e.g., Suzuki 2015 for L2 reading skill), and self-evaluation training/experience

(e.g., Birjandi and Bolghari 2015). Using Can-Do statements, Suzuki (2015) examined 63 Japanese learners' self-evaluations of Chinese reading skills. The correlation between their self-evaluations and the objective proficiency measures was not strong, varying from over- to underestimation. However, ID variables including the length of residence in Japan and the amount of reading experience contributed to the better calibration.

With regard to how IDs impact on the self-evaluation of L2 pronunciation and speech, similar tendencies can be observed in terms of L2 proficiency: learners with greater proficiency did not tend to overestimate their L2 speaking ability (Ma and Winke 2019; Roever and Powers 2005; Li 2018). For example, Li (2018) found that learners with greater listening and speaking skills tended to exhibit less overconfidence in their self-evaluations of comprehensibility. With regard to L2 language skills, the role of experience appears to be unclear. Although it may play a role in self-evaluation biases (e.g., Saito et al. 2020b), experience appears to have little impact in some cases (e.g., Isbell and Lee 2022). Saito et al. (2020b) found that increased practice, such as taking extra English classes, improved students' ability to judge their own comprehensibility. This suggests that enhanced English skills coupled with more practice can result in more accurate self-evaluations. Conversely, Isbell and Lee (2022) reported that experience did not significantly affect self-evaluations. Despite expectations that real-world language use would provide ample opportunities for learners to compare themselves to others and to receive feedback, this did not appear to influence the accuracy of their self-assessments. The authors speculated that learners with less precise self-evaluations may be less attuned to others' speech and the feedback they receive.

The focus on IDs in L2 self-evaluations has recently begun to shift toward learners' psychological factors, arguably due to the frequent observation of the Dunning–Kruger effect in self-assessments of L2 speech (Panadero et al. 2017; Isbell and Lee 2022): confident L2 learners tend to rate themselves higher than external evaluations of their performances would indicate. Isbell and Lee (2022) made one of the first attempts to explore the influence of psychological IDs on overconfident and underconfident evaluations made by L2 learners. In addition to the range of experiential variables, they investigated Korean learners' attitudes regarding the importance of English pronunciation (i.e., value) and their degree of satisfaction with their own pronunciation skills. The authors found that a learner's satisfaction with their pronunciation strongly influenced their self-evaluation of comprehensibility. In fact, those who placed a high value on pronunciation were prone to overestimating their comprehensibility. Similarly, satisfaction with pronunciation skills emerged as a significant predictor of self-assessments regarding accentedness, thus contributing to overconfidence and inaccurate self-judgments. While research has recently begun to adopt an ID paradigm in research on L2 self-evaluations, the existing studies have only paid attention to psychological and experiential factors and have neglected cognitive factors.

### 3. The Current Study

As evidenced by existing studies on self-evaluation, L2 learners often miscalibrate their language skills (e.g., Gaffney 2018). This trend can also be observed in L2 speaking (e.g., Dlaska and Krekeler 2008; Isbell and Lee 2022; Li 2018; Ortega et al. 2022). In particular, comparisons between self-evaluations and others' evaluations of L2 speech exhibit the Dunning–Kruger effect, with learners with low L2 oral ability overestimating their ability and high-level performers underestimating their ability (e.g., Ortega et al. 2022; Trofimovich et al. 2016). In order to understand the causes of this mismatch in more depth and to seek pedagogical applications for calibrating learners' perceptions, L2 scholars have begun to explore ID factors. Overall, one can classify the existing research on IDs as follows: biological (gender, age), experiential (e.g., length of residence, current amount of L2 use, total amount of L2 instruction received), and psychological/attitude factors (e.g., satisfaction with learners' pronunciation, the value they place on pronunciation). However, in the area of L2 self-evaluation, the influence of perceptual–cognitive abilities on the accuracy of self-assessments warrants further exploration.

With regard to the impact of IDs on L2 learning, scholars have identified a broad range of IDs, ranging from socio-psychological (e.g., motivation and emotion) to cognitive factors (e.g., foreign language aptitude, working memory, attention, and auditory processing). For L2 listening—a primary skill required for L2 learners to process, analyze, and evaluate themselves—linguistic knowledge (grammatical, phonological knowledge, and especially vocabulary knowledge, see Wallace (2022) for an example) serves as a primary component, while cognitive factors are believed to facilitate the rapid processing of speech (cf. Hulstijn 2019). To date, multifarious cognitive factors have been reported to be associated with better L2 listening proficiency, including working memory, phonological memory (e.g., Kormos and Sáfár 2008; Brunfaut and Révész 2015), implicit learning (Linck et al. 2013), and auditory processing (e.g., Vandergrift and Baker 2018; Kachlicka et al. 2019 for phonological perception). However, despite the influence of such cognitive IDs on L2 learners' processing of speech, these factors have not been considered in research on L2 self-evaluations.

The process of L2 self-evaluation also involves cognitive manipulation, which is similar to listening comprehension, namely, decoding linguistic information, integrating the processed information into long-term memory, and utilizing background knowledge. According to Hulstijn (2019), although linguistic knowledge is a primary factor that assists L2 learners in decoding and comprehending speech, cognitive IDs play a supporting role, particularly when learners lack sufficient linguistic knowledge. Based on this assumption, the influence of cognitive IDs on the accuracy of L2 speech self-evaluation was explored in the current study by adopting evidence from ID research on L2 listening to answer the following research questions (RQs):

RQ1—To what extent do cognitive IDs (i.e., working memory, phonological memory, implicit sequence learning, and auditory processing) influence self-evaluations of L2 speech? RQ2—How do cognitive IDs affectthe types of mismatches in self-evaluations?

With regard to RQ1, given the assumption that cognitive IDs can be a compensatory factor in listening, IDs that are believed to contribute to the efficient processing of acoustic information in L2 listening were included, namely, working memory, phonological memory, implicit sequence learning, and auditory processing (acuity and integration). Working memory is associated with L2 listening proficiency (Linck et al. 2013), and it has been proposed that the components of working memory impact on listening in various ways. For example, the storage component is essential for learning new vocabulary, while the processing components are important for real-time tasks such as listening (e.g., Wen et al. 2013). Since phonological memory is a key factor in processing auditory information, it is assumed to contribute to the efficient operations of speech processing and production (e.g., Baddeley 2000, 2003; Baddeley et al. 1998; Gagné et al. 2022; O'Brien et al. 2006) as well as to successful L2 learning (Kormos and Sáfár 2008). Therefore, in addition to the overall storage capacity, phonological memory has been included. Implicit sequence learning was included to assess the participants' implicit statistical learning ability, which is the ability to learn and identify phonological regularities. This ability is thought to be the basis for learning and identifying new sound strings (e.g., word forms) in the incoming auditory information (e.g., Gómez and Gerken 2000; Jusczyk 1997; Saffran et al. 1996). In the context of listening, this ability will assist L2 learners to segment speech into discrete word units, to identify the lexical units in a language, and to develop automaticity in their speech processing (Speciale et al. 2004), thus contributing to the rapid decoding of incoming L2 speech. In fact, Linck et al. (2013) found a strong association between sequence learning ability (measured via serial reaction time task) and high-level L2 listening proficiency. With respect to auditory processing ability, past research indicates a positive relationship between sound discrimination ability and L2 listening proficiency (e.g., Wilson et al. 2011; also see Saito et al. 2023; Vandergrift and Baker 2015). Auditory processing is the first ability that listeners use to perceive incoming acoustic signals; they then transfer the information for further cognitive manipulation (i.e., working memory). Hence, it is considered to be an essential ability in the bottom-up process of accurate L2 listening. Based on this, the following hypothesis was proposed: due to the secondary role of cognitive factors

in L2 comprehension (Hulstijn 2015, 2019), the contribution of cognitive IDs may not be as significant as that of learners' linguistic knowledge or L2 learning-related experiences. However, they would influence the accurate self-evaluation of L2 speech regardless of dimensions such as comprehensibility (i.e., ease of understanding) or accentedness (i.e., the degree of a perceived foreign accent). A strong prediction cannot be made for RQ2 due to the lack of prior research. Therefore, any research findings should be considered to be exploratory.

## 4. Materials and Methods

### 4.1. Participants

A total of 97 Japanese learners of English in Japan participated in the current study (M age = 20.1 years; Range$_{age}$ = 18–26 years). At the time of the project, the participants had received equal amounts of English education in junior high and high school, and were enrolled in various undergraduate courses at universities in Japan. Their proficiency in English varied widely, ranging from A2 to C1 on the Common European Framework of Reference for Languages based on their Test of English for International Communication (TOEIC) test score.[2] None of the participants reported difficulty with hearing or having difficulty reading.

### 4.2. Procedure

The data collection took place online using the online data collection platform Gorilla (Anwyl-Irvine et al. 2020). After completing a set of consent forms, the participants were asked to use their PCs to log into the experiment, and completed a range of tasks in the following order: linguistic and cognitive tasks, a speaking task, a self-evaluation task, and an experience questionnaire that elicited participants' information related to L2 learning and use. The entire session lasted for approximately 60 min. The descriptive statistics for all the ID and self-evaluation scores are presented in Appendix A. The researcher used Gorilla's progress tracking function, which displays the timing of task completion, to monitor the participants' compliance with the instructions and their performances remotely.

### 4.3. Instruments

Since the construct of comprehensibility includes not only phonological elements but also lexicogrammatical accuracy, linguistic knowledge is assumed to be the primary source of L2 speech comprehension. Therefore, the participants' grammatical and lexical knowledge was measured. In addition, to determine participants' perceptual–cognitive IDs, which are believed to assist with L2 speech comprehension and processing, they completed a digit span task, a non-word serial recognition task, serial reaction time tasks, and a series of auditory discrimination and memory tasks.

#### 4.3.1. Grammaticality Judgment Task

To gauge the participants' relatively automatized yet explicit grammatical knowledge (e.g., Suzuki and DeKeyser 2017), a timed grammaticality judgment task (GJT) was used in the current study (for a similar methodological decision, see Kachlicka et al. 2019). In the GJT, which was first introduced by Godfroid and colleagues (Godfroid et al. 2015), the participants read and evaluate the grammaticality of 68 sentences. These sentences vary in length (five to 12 words) and target structures, such as plurality, tense, and articles. Of these sentences, half are grammatically correct, while the remaining sentences contain morphosyntactic errors. The sentences are presented randomly, and participants judge them one at a time. Each sentence has a unique response time limit, ranging from 1800 to 6240 milliseconds based on the estimated processing time (for detailed calculations, see Godfroid et al. 2015). Accuracy ratio scores, ranging from 0 to 100%, are then generated automatically to reflect the participants' ability to identify grammatical sentences correctly and to reject ungrammatical sentences.

### 4.3.2. L2 Vocabulary Knowledge Test

L2 vocabulary knowledge is another crucial component of linguistic ability that L2 learners need in order to process L2 speech. Therefore, LexTALE (Lemhöfer and Broersma 2012) was used to measure the participants' lexical competence via an untimed lexical decision task consisting of 60 items in total, including 40 real words and 20 non-words. The participants were asked to judge whether the presented item is a real word or a non-word. The score was computed using correct identification ratios ranging from 0 to 100%.

### 4.3.3. Working Memory

The digit span task consisted of two segments, namely, a forward span and a backward span. In the forward span condition, the participants viewed a sequence of numbers, each of which was displayed for 500 milliseconds, and were then tasked with remembering them in the presented order. In the backward span condition, the participants were asked to recall and input the numbers in the reverse sequence on their computers. Each number series varied in length and contained 3 to 11 digits, with two attempts being allowed for each sequence length. The participants' scores were based on the maximum number of digits that they recalled accurately in both attempts for each segment. The average scores from both spans were calculated to determine the participants' overall working memory scores.

### 4.3.4. Phonological Memory

Phonological memory was measured via a serial nonword recognition task (e.g., Gathercole et al. 2001; Isaacs and Trofimovich 2011; O'Brien et al. 2006). The participants listened to 24 pairs of one-syllable non-word sequences (i.e., consonant–verb–consonant nonwords) that varied in length from five to seven words (eight pairs of five-word sequences, eight pairs of six-word sequences, and eight pairs of seven-word sequences). The paired sequences were categorized according to two patterns of "same" and "different". In the "same" pattern, a sequence was repeated with a brief pause in between, while in the "different" pattern, the second sequence in a pair had two adjacent non-words switched. To ensure that the listeners processed the entire sequence and not simply the transposed items, neither the first nor the last item in a sequence was ever swapped. The positions of these swapped items changed randomly across the different sequence lengths. Each non-word was separated by about 0.8 s, and a pause of 1.5 s between sequences was given. The participants' task was to listen to the pairs and to then decide whether the paired sequences were the same or were different.

### 4.3.5. Implicit Learning

Two types of serial reaction time tasks (visual and auditory) were used to assess sequence learning (Linck et al. 2013; Tagarelli et al. 2016; Willingham et al. 1989). In the visual mode, four boxes were displayed on the screen, with each box being a potential location for an asterisk. During each trial, an asterisk appeared in one of these boxes, prompting the participants to press the corresponding button on a response device. After a pause of 0.5 s, the asterisk appeared in a new location. In order to help the participants to become familiar with the task format, a practice session was provided prior to the main task.

The test consisted of six blocks, each with 60 trials. Only the first and the last blocks had asterisks appearing in a pseudorandom sequence. For the second to the fifth blocks, the asterisks followed a 10-item repeating pattern. The test produced an implicit sequence learning score, which was determined by the differences in response times between the final sequential and random blocks. A higher score indicated better sequence learning.

In the auditory mode, the design of the task was similar to that in the visual mode except for the stimuli and how the participants reacted to them. The stimuli were four distinguishable notes using the first five notes of the major scale (i.e., 220, 246.9, 277.2, and 311.1 Hz), and each note corresponded to a particular key on the keyboard. In the practice phase, identification practice was included to ensure that the participants could match a

particular note to its corresponding key. After the practice phase, the participants proceeded to the main task in which the first and last blocks followed pseudorandom sequences, while the second to fifth blocks repeated a specific sequence. The implicit learning score was calculated in the same manner as in the visual mode: the scores for the two modes of the tasks were standardized and averaged to produce an implicit learning score.

### 4.3.6. Perceptual Acuity

The participants engaged in four AXB discrimination tasks tailored to assess their ability to detect minimal sound differences. Each subtest focused on one acoustic dimension—formants, duration, risetime, and pitch. In each trial, they were provided with three sounds and asked to detect the one that is different from the other two. Following Saito et al. (2023), each score for the subtests was averaged to produce an overall auditory acuity score for each participant. The lower the score, the better the participants' ability to encode more detailed acoustic characteristics. For a more detailed description of the task design, see Kachlicka et al. (2019).

### 4.3.7. Audio-Motor Integration

In the current study, the participants undertook two audio-motor integration tasks, namely, melody and rhythm reproduction. In the melody test, they listened to 10 sequences of melodies with combinations of seven notes in each, and then reproduced each melody by pressing buttons corresponding to the notes on their keyboards. The participant's performances were scored based on the accuracy of the selected notes. In the rhythm task, the participants listened to patterns of drum beats, then mimicked the timing by pressing a space key. Their timing accuracy determined their score. Both tasks aimed to test the participants' adaptability to new motor tasks without prior practice, thus minimizing the reliance on memory. Please see Tierney and colleagues (Tierney et al. 2017) for details of the task designs.

### 4.4. Speaking and Self-Evaluation Task

After submitting the consent forms, the participants proceeded to the speaking and evaluation phases. Following the existing L2 speech evaluation paradigm (e.g., Trofimovich et al. 2016) in which semi-spontaneous speech is used as the indicator of L2 learners' oral ability as well as to elicit a sufficient amount of speech even from the learners with low proficiency, a picture-description task was adapted from the EIKEN English Test (EIKEN Foundation of Japan 2016; also see Lambert et al. 2017). Here, the pictures depict a story of a girl who receives a smartphone for children by convincing her parents to buy one for her. After the microphone test and a practice task, the participants were given 60 s of planning time and another 60 s to produce a response. Immediately after the speaking task, they proceeded to the evaluation phase in which they were briefed about the evaluation scales and procedures. As has been done in a range of existing studies on L2 speech (e.g., Derwing and Munro 2013; Nagle 2018) and self-evaluations (Saito et al. 2020b), we operationalized intuitive judgments through scaler judgments of overall comprehensibility and accentedness.

While some studies ask participants to recall their previous performance or guess their proficiency without any reference/source and estimate how well they did in speaking tasks (e.g., Trofimovich et al. 2016), the current study used participants' own recording as a source of self-evaluation (see also Li 2018; Strachan et al. 2019; Tsunemoto et al. 2022). On the evaluation screen, a button and two Likert scales were displayed. The task was to press the button to play the audio recording of what they had just performed in the speaking phase, then assess their speech based on the provided criteria—comprehensibility and accentedness. For making smooth judgements, a 9-point Likert scale along with the explanation of evaluation criteria were displayed on the same experiment screen. Comprehensibility was introduced as the degree of effort required in understanding the speech. Accentedness was described as the extent to which the speech was influenced by

their L1. The participants were allowed to play the recording as many times as they liked to arrive at their satisfactory evaluation scores.

### 4.4.1. L2 Speech Evaluation

Six native English speakers (three females and three males) assessed speech samples for comprehensibility and accentedness using a 9-point Likert scale (for similar decisions, see Derwing and Munro 2013; Nagle 2018). In order to minimize the influence of listener factors (e.g., Kennedy and Trofimovich 2008 for language teaching experience; Winke et al. 2013 for familiarity with accents), we controlled for their backgrounds, teaching experience, and accent familiarity. The six raters had Master's or PhD degrees in applied linguistics and related fields, and had experience of teaching English to L2 learners (M = 4.2 years). In addition, special attention was paid to their familiarity with Japanese-accented English to ensure consistent evaluation sensitivity. Following the existing studies of speech rating that employed impressionistic judgments (e.g., Suzuki and Kormos 2019), a 6-point Likert scale was used to measure accent familiarity in the current study. The raters reported relatively high familiarity with this accent (M = 5.1; Range = 4–6), thus ensuring consistent sensitivity to Japanese-accented English. No rater reported having any hearing impairments.

### 4.4.2. Procedure for the Speech Evaluation

The rating session was conducted online. Prior to the main session, the raters completed the training session that provided them with a detailed explanation of the procedure, as well as the definitions of comprehensibility and accentedness. This was followed by practice rating using three speech samples that were not included in the main dataset. In both the practice and the main rating sessions, the raters were asked to listen to speech samples through headphones connected to a laptop computer, and to subsequently evaluate the samples by clicking a number on a nine-point scale to indicate the degree of accentedness and comprehensibility on a rating screen. The raters then proceeded to the main session. To avoid fatigue, the raters took breaks of two to three minutes after one third and two thirds of the speech samples had been evaluated. The entire session lasted approximately 65 min for each rater.

The Cronbach's alphas for the six raters' judgments of comprehensibility and accentedness were $\alpha = 0.85$ and $\alpha = 0.81$, respectively. Since the Cronbach alphas reached the acceptable value suggested by Larson-Hall's (2010) benchmark ($\alpha > 0.70$), the results of the judgments were averaged to represent each speaker's comprehensibility and accentedness scores. In addition, the speakers' comprehensibility scores were used as one of the indices for their linguistic ability: overall speaking proficiency. The descriptive statistics for the rating results are summarized in Appendix A.

### 4.5. Data Analysis

Two types of self-evaluation scores were calculated to answer the RQs, namely, a confidence score and a distance score. The confidence score indicated the nature of the discrepancies between the self-assessments and others' evaluations. Following prior research (e.g., Trofimovich et al. 2016), the confidence score was calculated by subtracting the mean raters' scores from the participants' self-assessment scores. The positive scores indicated that the participants were overconfident about their speech, while the negative scores showed that the participants were underconfident about their speech performances (also see Saito et al. 2020b). For example, if a participant's comprehensibility score was 4 and their self-evaluated comprehensibility score was 6, their confidence score was 2. This means that the participant was overconfident about their L2 speech comprehensibility. If a participant's comprehensibility score was 4 and their self-evaluated comprehensibility score was 2, their confidence score was −2, indicating that the participant was underconfident about their self-evaluated comprehensibility. The participants' confidence scores were computed for comprehensibility and accentedness, and were grouped according to overconfident and underconfident learners. A *t*-test was computed to verify the accuracy

of the grouping statistically. Welch's *t*-test was selected due to the unbalanced sample sizes (32 vs. 65). The result indicates that the confidence scores for the two groups showed a significant effect of group (t = 11.47, *p* < 0.001, Cohen's d = 2.37), thus suggesting that the differences in the self-evaluation scores for each group were statistically significant.

Although the confidence scores indicate the difference between the native speakers' ratings and the participants' self-evaluations, they do not reflect the amount of inaccuracy regardless of overconfidence or underconfidence. The primary aim of RQ1 was to identify the influence of IDs on the inaccuracies in the self-evaluations. Therefore, to quantify how far the self-evaluations were from the baseline native speakers' evaluation scores, the values of the confidence scores were converted into absolute values; thus, the distance from the native baseline could be quantified as a positive value regardless of the type of confidence. These distance scores were computed for the self-evaluations of comprehensibility and accentedness. The descriptive statistics for the confidence score, the distance score, and the ID variables are presented in Appendix A. In addition, since the results of the Shapiro–Wilk tests suggest that the *p*-values for the variables were smaller than 0.05, except for grammar knowledge, auditory acuity, and auditory reproduction, the decision to use Spearman's rho to run a correlation analysis was made. The results are summarized in Appendix A.

The participants' distance scores for comprehensibility and accentedness were used to answer RQ1. Using the IDs as fixed effects, a series of mixed-effect models were computed to identify the best model fit for both constructs. These analyses were performed via the lmer functions from the lme4 package (Version 4.2.2; Bates et al. 2015) in the R statistical environment (R Core Team 2022). The Akaike Information Criterion (AIC) was used for the model comparisons. A lower AIC value generally indicates a better-fitting model. With regard to RQ2, the participants' confidence scores (i.e., the scores for overconfidence and underconfidence) were used to run multiple regression analyses to identify how IDs affected the types of miscalibrations (overconfidence or underconfidence) and the relative weights of these variables in the full model by running dominance analyses (Mizumoto 2023).

## 5. Results

### 5.1. Discrepancies between Self-Evaluations and Others' Evaluations

A series of correlation analyses was conducted first to examine the discrepancies between self-evaluations and others' evaluations of L2 speech comprehensibility and accentedness (see Appendix A for the correlation results). According to the correlation results, the correlations between the participants' self-evaluations and L1 English listeners' evaluations were weak in terms of both comprehensibility (r = 0.21, *p* = 0.039) and accentedness (r = 0.20, *p* = 0.064). These results suggest that the participants and the L1 listeners in the current study do not align well in terms of speech evaluation. With regard to the relationship between the participants' confidence scores and their actual scores (i.e., English listeners' evaluation), both comprehensibility and accentedness showed strong negative correlations (r = −66, *p* < 0.001 for comprehensibility; r = −65, *p* < 0.001 for accentedness). Such strong negative correlations indicate that the poorer the L2 pronunciation performance is, the higher the participants tend to self-assess themselves. Furthermore, the correlations between the participants' distance scores and L1 English listeners' evaluations of comprehensibility were moderate but statistically significant (r = 0.29, *p* = 0.004), while the correlation coefficient was weak and statistically non-significant in terms of accentedness (r = −0.13, *p* = 0.210). These results suggest that participants who showed inaccurate self-assessment (i.e., overconfidence or underconfidence) tended to demonstrate higher comprehensibility, but spoke with a heavier degree of L1 accent.

### 5.2. Contribution of Linguistic Knowledge and IDs to Accurate Self-Evaluation

The first objective of the statistical analysis was to examine the extent to which cognitive IDs could explain the inaccuracy of L2 speech self-evaluation. To this end, model comparisons were carried out by constructing possible models through the mixed-effect modeling analyses. The analysis was performed via the lmer functions from the lme4

package (Version 4.2.2; Bates et al. 2015) in the R statistical environment (R Core Team 2022). Based on the inspection of correlation results, none of the dependent variables were strongly correlated with each other (see Appendix A for the correlation results). Since multicollinearity was not detected, L2 linguistic proficiency (grammar, vocabulary, and pronunciation), experience (age of learning, hours of English classes per week, experience of having studied abroad, and hours of conversation in English per week), and cognitive IDs (perceptual acuity, audio-motor integration, working memory, phonological memory, implicit learning) were included as fixed effects. Group, which is a categorical variable based on the distinction between overconfident and underconfident speakers, was included as a random effect.

First, a null model was constructed with a random effect of Group (see Table 1 for R codes). Model 1 included L2 linguistic proficiency variables, Model 2 included experience variables, and Model 3 included cognitive IDs. According to the ANOVA comparisons of the models, the AIC value for Model 1 (AIC = 323.01) was smaller than it was in the null model (AIC = 326.3), while the values for Model 2 (AIC = 332.59) and Model 3 (AIC = 328.95) were higher than the value for the null model (AIC = 332.59). The chi-square value indicates that Model 1 (i.e., the inclusion of L2 proficiency variables) was a much better fit compared to the rest of the models ($x^2$ = 8.35, *p* = 0.039). Since the model with experiential and cognitive variables did not improve in terms of model fit (Model 2), Model 1 was considered to be the best fit. The variances explained by the L2 proficiency variables were 4.1% ($R^2_{marginal}$ = 0.041), and 71% with a random effect ($R^2_{conditional}$ = 0.71). Details of the final model (Model 1) are presented in Table 2.

**Table 1.** R codes used for the mixed effect modeling analyses.

| Model | Code |
|---|---|
| Null model | Distance score~(1 \| Group) |
| Model 1: | Distance score~L2 speech proficiency + Vocabulary knowlege + Grammarknowlege + (1 \| Group) |
| Model 2: | Distance score~English classes + Living abroad experience + Conversation frequency + Age of learning + (1 \| Group) |
| Model 3: | Distance score~Working memory + Phonological memory + Perceptual acuity + Audio-Motor integration + Implicit learning + (1 \| Group) |
| Model 4: | Distance score~L2 speech proficiency + Vocabulary knowlege + Grammar knowlege + Working memory + Phonological memory + Perceptual acuity + Audio-Motor integration + Implicit learning + English classes + Living abroad experience + Conversation frequency + Age of learning + (1 \| Group) |

With regard to the self-evaluations of accentedness, model comparisons were carried out in the same manner as those for comprehensibility. The null model was compared to Model 1, Model 2, and Model 3 (see Table 1). The result of the comparisons reveal that none of the variables improved the model fit from the null model (AIC = 292.96 for the null model; AIC = 297.47 for Model 1; AIC = 297.88 for Model 2) except for Model 3 (AIC = 291.97). The comparison between the null model and Model 3 almost reached statistical significance ($x^2$ = 10.99, *p* = 0.051). The full model (Model 4) did not show improved model fit either (AIC = 752.14). While the ANOVA comparison showed that Model 3's statistical significance was marginal, the AIC value was lower in Model 3 than it was in the null model. Therefore, Model 3 was selected as the final model for accentedness. The variances explained by this model were 3.1% ($R^2_{marginal}$ = 0.031), and 74% with a random effect ($R^2_{conditional}$ = 0.74). Details of the final model (Model 3) are presented in Table 3.

**Table 2.** The final model for self-evaluation distance score (Comprehensibility).

|  | Estimate | SE | df | *t* Value | *p* |
|---|---|---|---|---|---|
| (Intercept) | 3.63 | 1.3 | 7.41 | 2.79 | 0.026 |
| Perceptual acuity | −0.002 | 0.007 | 89.4 | −0.296 | 0.768 |
| Audio-motor integration | −0.014 | 0.02 | 89.2 | −0.707 | 0.481 |
| Working memory | 0.122 | 0.126 | 89.1 | 0.97 | 0.335 |
| Phonological memory | −0.209 | 0.106 | 89 | −1.97 | 0.052 [†] |
| Implicit learning | 0.345 | 0.14 | 89.1 | 2.46 | 0.016 * |
|  | **Variance** | **SD** |  |  |  |
| Group | 2.57 | 1.60 |  |  |  |
| Residual | 0.945 | 0.972 |  |  |  |
| Marginal $R^2$ | 0.031 |  |  |  |  |
| Conditional $R^2$ | 0.739 |  |  |  |  |

Note that, for Perceptual acuity, lower scores indicate better discrimination ability. * indicates statistical significance ($p < 0.05$), [†] indicates marginal significance ($p < 0.10$).

**Table 3.** The final model for self-evaluation distance score (Accentedness).

|  | Estimate (b) | SE | df | *t* Value | *p* |
|---|---|---|---|---|---|
| (Intercept) | 2.02 | 1.08 | 2.55 | 1.88 | 0.174 |
| Grammar knowledge | −0.001 | 0.120 | 91.1 | −0.009 | 0.993 |
| Vocabulary knowledge | −0.029 | 0.123 | 91.3 | −0.235 | 0.815 |
| Speech proficiency | 0.020 | 0.007 | 92.8 | 3.03 | 0.003 * |
|  | **Variance** | **SD** |  |  |  |
| Group | 3.03 | 1.74 |  |  |  |
| Residual | 1.33 | 1.16 |  |  |  |
| Marginal $R^2$ | 0.059 |  |  |  |  |
| Conditional $R^2$ | 0.749 |  |  |  |  |

Note that * indicates statistical significance ($p < 0.05$).

### 5.3. The Effect of Cognitive IDs on the Type of Self-Evaluation Biases

A series of multiple regression analyses was performed to identify the role of cognitive IDs in the overconfident and underconfident participants' self-estimations of speech quality. First, to meet the regression assumptions and to avoid multicollinearity issues among the variables, the variance inflation factors (VIF) were inspected. The results reveal that the values were less than 3; thus, the multicollinearity problem was not a factor (Winter 2019).

The participants' confidence scores for comprehensibility and accentedness (i.e., the values below 0 indicate underconfidence and the values above 0 indicate overconfidence) were used as dependent variables in the analyses. The participants' experiential and cognitive IDs, as well as the L2 proficiency variables, were submitted to the regression models. With regard to the confidence score for comprehensibility, L2 speech proficiency ($\beta = -0.800$, t = $-8.27$, $p < 0.001$) and implicit learning ($\beta = 0.544$, t = 1.91, $p = 0.059$) were found to be statistically significant and marginally significant predictors (see Table 4). The dominance analysis revealed that a major part of the weight was occupied by L2 speech proficiency (73.4%), while a small amount was explained by implicit learning (9.6%). With regard to the direction of the IDs' influence on the confidence score (i.e., the focus of this analysis), while most of the experience-related variables (age of learning and number conversations in English) contributed to overconfidence, most of the cognitive IDs (e.g., working memory, audio-motor integration, and phonological memory), L2 proficiency variables (e.g., grammar knowledge) and experience of having studied abroad appeared to have a negative effect on the confidence scores. Compared to the rest of the cognitive IDs, perceptual acuity and implicit learning had an influence in the opposite direction; that is, better scores for those abilities were associated with scores for overconfidence.

**Table 4.** Multiple regression result of comprehensibility confidence score.

| | Estimate (b) | SE | t Value | p | Relative Weight | |
| | | | | | Raw Weight | Rescaled Weight |
|---|---|---|---|---|---|---|
| (Intercept) | 5.38 | 2.13 | 2.52 | 0.014 * | | |
| Age of learning | 0.043 | 0.068 | 0.622 | 0.536 | 0.001 | 0.087% |
| English classes per week | 0.003 | 0.083 | 0.035 | 0.972 | 0.002 | 0.328% |
| Study abroad experience | −0.664 | 0.433 | −1.54 | 0.129 | 0.028 | 5.38% |
| Conversation per week | 0.110 | 0.084 | 1.31 | 0.195 | 0.012 | 2.24% |
| Grammar knowledge | −0.147 | 0.207 | −0.71 | 0.480 | 0.005 | 0.856% |
| Vocabulary knowledge | 0.288 | 0.205 | 1.41 | 0.163 | 0.009 | 1.72% |
| Speech proficiency | −0.800 | 0.097 | −8.27 | <0.001 * | 0.387 | 73.4% |
| Perceptual acuity [a] | −0.009 | 0.013 | −0.710 | 0.479 | 0.017 | 3.23% |
| Audio-motor integration | −0.028 | 0.039 | −0.718 | 0.475 | 0.003 | 0.539% |
| Working memory | −0.290 | 0.251 | −1.15 | 0.252 | 0.009 | 1.62% |
| Phonological memory | −0.068 | 0.213 | −0.320 | 0.750 | 0.002 | 0.336% |
| Implicit learning | 0.544 | 0.285 | 1.91 | 0.059 [†] | 0.049 | 9.59% |

Note that for Perceptual acuity [a], lower scores indicate better discrimination ability. * indicates statistical significance ($p < 0.05$), [†] indicates marginal significance ($p < 0.10$).

In terms of accentedness, the multiple regression analysis showed that none of the variables were statistically significant predictors of the confidence score for accentedness (see Table 5). Unlike comprehensibility, the dominance analysis revealed that the weight occupied by L2 speech proficiency was smaller (17.7%), and that implicit learning accounted for a large proportion of the weight (23.7%). Concerning the direction of the IDs' influence on the confidence score, the majority of the experiential IDs (age of learning, conversation in English, amount of conversation per week, and experience of having studied abroad), two cognitive IDs (audio-motor integration and phonological memory) and L2 proficiency variables (grammar knowledge and speech proficiency) were found to contribute to underconfident estimations of accentedness, whereas three cognitive IDs (perceptual acuity, working memory, and implicit learning) and vocabulary knowledge played a role in the overestimated accentedness scores. A summary of the directions in which the variables influenced the participants' self-evaluations is presented in Table 6.

**Table 5.** Multiple regression result of accentedness confidence score.

| Accentedness | Estimate (b) | SE | t Value | p | Relative Weight | |
| | | | | | Raw Weight | Rescaled Weight |
|---|---|---|---|---|---|---|
| (Intercept) | 6.51 | 2.80 | 2.33 | 0.022 * | | |
| Age of learning | −0.043 | 0.089 | −0.487 | 0.628 | 0.001 | 0.761% |
| English classes per week | −0.111 | 0.108 | −1.02 | 0.311 | 0.010 | 9.16% |
| Study abroad experience | −0.915 | 0.567 | −1.61 | 0.111 | 0.023 | 20.8% |
| Conversation per week | −0.077 | 0.110 | −0.697 | 0.488 | 0.003 | 3.06% |
| Grammar knowledge | −0.114 | 0.271 | −0.42 | 0.676 | 0.002 | 1.65% |
| Vocabulary knowledge | 0.179 | 0.268 | 0.666 | 0.507 | 0.007 | 6.8% |
| Speech proficiency | −0.090 | 0.127 | −0.712 | 0.479 | 0.019 | 17.7% |
| Perceptual acuity [a] | −0.024 | 0.017 | −1.44 | 0.154 | 0.002 | 2.06% |
| Audio-motor integration | −0.060 | 0.051 | −1.16 | 0.250 | 0.010 | 9.54% |
| Working memory | 0.038 | 0.330 | 0.115 | 0.909 | 0.001 | 0.498% |
| Phonological memory | −0.227 | 0.279 | −0.813 | 0.419 | 0.005 | 4.28% |
| Implicit learning | 0.256 | 0.374 | 0.684 | 0.496 | 0.026 | 23.7% |

Note that for perceptual acuity [a], lower scores indicate better discrimination ability. * indicates statistical significance ($p < 0.05$).

**Table 6.** Summary of linguistic, experiential and cognitive variables' influences on self-evaluation.

| Variables | Comprehensibility | Accentedness |
|---|---|---|
| Age of Learning | overconfidence | underconfidence |
| English classes per week | underconfidence | underconfidence |
| Study abroad experience | overconfidence | overconfidence |
| Conversation per week | overconfidence | underconfidence |
| Grammar knowledge | underconfidence | underconfidence |
| Vocabulary knowledge | overconfidence | overconfidence |
| Speech proficiency | underconfidence | underconfidence |
| Perceptual acuity | overconfidence | overconfidence |
| Audio-motor integration | underconfidence | underconfidence |
| Working memory | underconfidence | overconfidence |
| Phonological memory | underconfidence | underconfidence |
| Implicit learning | overconfidence | overconfidence |

## 6. Discussion

### 6.1. Self-Evaluation Accuracy

In order to fill the research gap, the current study examined (a) the influence of cognitive IDs on the accuracy of L2 speech self-evaluation (i.e., RQ1), and (b) how cognitive IDs affect the type of mismatch in self-evaluation (i.e., RQ2). According to the results, the participants' self-evaluations and L1 English listeners' evaluations of comprehensibility were weakly correlated, and the correlation was also weak in terms of accentedness. Judging from the existing L2 speech self-evaluation studies, which yielded mixed results in terms of comprehensibility (Trofimovich et al. 2016; Li 2018; Strachan et al. 2019; Saito et al. 2020b; Isbell and Lee 2022) and accentedness (Trofimovich et al. 2016; Li 2018; Isbell and Lee 2022), the current study supports the findings reported by Trofimovich et al. (2016) and Saito et al. (2020b), that L2 learners do not accurately perceive their own comprehensibility and accentedness. Moreover, the current study demonstrated that even when they can listen to their own speech, this mismatch still seems to occur. When comparing the correlation coefficient of comprehensibility and accentedness, it appears to be the case that assessing comprehensibility is somewhat more challenging than accentedness for L2 learners.

With regard to the relationship between the participants' confidence scores (i.e., how much higher/lower a learner's self-assessment was compared to a listener-based assessment of that learner) and their actual scores (i.e., English listeners' evaluations), both comprehensibility and accentedness had strong negative correlations. The findings corroborate the results of previous studies that have identified a strong negative link between confidence scores and others' evaluations (e.g., Saito et al. 2020b), indicating that L2 learners who were rated as being more comprehensible and less accented by listeners often underestimated their own speech, whereas those who were perceived less favorably by others had a tendency to overestimate their own capabilities. This finding further confirms the presence of the Dunning–Kruger effect in L2 speech evaluations (cf. Kruger and Dunning 1999; Li and Zhang 2021; Ross 1998). Turning the focus to the distance scores (i.e., absolute differences between self-assessments and native speakers' assessments), the current study reveals that the correlations between the distance scores and the native speakers' evaluations differed depending on the constructs. While there did not appear to be an association in terms of accentedness, the result shows a strong negative association in terms of comprehensibility; that is, more comprehensible L2 learners tended to make self-assessments that differed from the assessments that the listeners made. This finding is different from that of Isbell and Lee's (2022) study, which found that more comprehensible speakers could self-assess their comprehensibility accurately. A follow-up Welch's *t*-test analysis, which compared the absolute distances of the discrepancies between the overconfident group and the underconfident group, suggested that the underconfident group was further from the L1 English listeners' evaluations (M = 2.5, SD = 1.9) than the overconfident group was (M = 1.4, SD = 1.4). This result further suggests that the participants tended to perceive

their speech to be poor and less comprehensible even though they were actually reasonably comprehensible.

*6.2. The Role of IDs in the Miscalibration of L2 Speech Evaluation*

With regard to the first research question, which explored the role of IDs in the miscalibration of L2 speech evaluation, a hypothesis based on Hulstijn's (2015, 2019) claim that cognitive factors played a minor role in comparison to linguistic knowledge in L2 listening comprehension was formulated for the current study. Since self-evaluation also involves listening and analyzing one's own speech, it was hypothesized that, while the impact of cognitive IDs might not be as significant as the effect of the learners' linguistic knowledge or their experience IDs, these cognitive factors would still have an influence on the accuracy of the learners' self-evaluations of their L2 speech. The mixed-effect model comparisons reveal that, among L2 linguistic proficiency (grammar, vocabulary, and pronunciation), experience (age of learning, hours of English classes per week, study abroad experience, and hours of conversation in English per week), and cognitive IDs (perceptual acuity, audio-motor integration, working memory, phonological memory, and implicit learning), the distance scores for comprehensibility could be best explained by a model with L2 linguistic proficiency variables, whereas accentedness was explained by cognitive IDs. Overall, such a link may be explained by the differences in the constructs of comprehensibility and accentedness. L2 speech research shows that listeners' attention is drawn to various linguistic aspects of speech, ranging from pronunciation to lexical and grammatical accuracy when judging comprehensibility, while pronunciation alone tends to explain the judgment of accentedness (see Crowther et al. 2015; Kang et al. 2010; Saito et al. 2016, for instance). According to studies that explore the roles of cognitive IDs in L2 learning, explicit and implicit learning aptitude may contribute to improving learners' representation of L2 systems and their processing abilities (e.g., Abrahamsson and Hyltenstam 2008; Granena 2013). Such tendencies appear to be the case in terms of L2 pronunciation. For instance, learners with better aptitude profiles have demonstrated superior segmental and prosodic perception (Kachlicka et al. 2019; Saito et al. 2020a) and production (e.g., Saito et al. 2019). In the case of the current study, phonological memory (i.e., the ability to retain phonological information for more thorough sound decoding) was found to have a primary influence on accurate calibration. Therefore, the current study corroborates such evidence, suggesting that cognitive IDs may play a role in the self-evaluation of accentedness, which appears to require fine-grained segmental and phonological analysis to detect the influence of one's own L1 on L2 performance. In contrast, cognitive IDs did not appear to significantly influence the self-evaluation of comprehensibility, as this construct is considered to encompass factors beyond pronunciation components (Saito et al. 2016).

With respect to implicit learning, it was found that the participants with better implicit learning tended to miscaliburate their accentedness. Implicit learning of sequence (internalization of language patterns without conscious learning) may have led to inaccurate accentedness evaluation due to their familiarity with their own speech. Learners who are frequently exposed to their own speech patterns seem to develop familiarity with their voice features, resulting in perceiving it to be highly intelligible (Mitterer et al. 2020). In the EFL contexts (the context wherein the participants are), where exposure to the target language is often limited to the classrooms, L2 learners who excel at implicit learning are continuously exposed to and practice their own speech patterns, rather than hearing speech produced by other users of English. This learning condition could lead to a greater familiarity with their own specific manner of speaking, instead of internalizing target language patterns through others' speech.

In the case of comprehensibility distance scores (i.e., the amount of gap between self- and other-assessment scores), speech proficiency (i.e., the participants' actual comprehensibility) was found to contribute significantly to the miscalibration. This means that the higher the speech proficiency, the poorer their estimation was. As reported in the previous section, the participants' self-assessment of speech exhibited the Dunning–Kruger effect

(also see Saito et al. 2020b). Therefore, the current result may have been observed because the proficient participants wrongly underestimated their comprehensibility level. In turn, the finding could suggest that compared to accentedness, where they can concentrate on self-assessing the phonological features of their own speech, comprehensibility judgement may be more susceptible to the Dunning–Kruger effect.

Unlike L2 speech proficiency, their grammar and vocabulary knowledge appeared to contribute to a smaller distance score (i.e., better calibration). While the implementation of the comprehensibility judgment requires listeners to make a holistic and impressionistic judgment on the degree of easiness of understanding (or the amount of effort required to understand the speech), those who have better grammar and vocabulary knowledge could pay attention to the details of how well they could use grammatical and lexical items to deliver intended meanings. Due to the crucial role of lexicogrammatical accuracy in achieving better comprehensibility (cf. Saito et al. 2016), the participants may have needed to have a high level of grammar and vocabulary knowledge to globally and accurately assess how well they could convey the message they wanted to deliver.

### 6.3. Impact of IDs on the Overconfidence and Underconfidence in Self-Evaluation

Concerning the second research question, an exploratory approach was adopted in the current study to examine which learner IDs contributed to overconfidence and underconfidence in the self-evaluations. On one hand, the multiple regression analyses revealed that experience of having studied abroad, vocabulary knowledge, perceptual acuity, and implicit learning were associated with the overestimation of comprehensibility and accentedness. On the other hand, the number of English classes per week, grammar knowledge, speech proficiency, audio-motor integration, and phonological memory were linked to underestimation. Such patterns can be speculated upon from various cognitive and experiential perspectives. For example, L2 learners with experience of having studied abroad may have overestimated their skills due to immersion and intensive interactions in the target language environment, which might have led to inflated perceptions of their linguistic abilities. Enhanced vocabulary knowledge may have contributed to overconfidence because these learners may have felt more equipped to understand and make themselves understood, and may have overlooked the finer nuances of language use that still needed improvement. Furthermore, perceptual acuity may have fostered overconfidence because the learners with refined auditory discrimination could have mistaken their ability to detect subtle phonetic differences for greater proficiency in producing those sounds. This sensitivity to nuances of sound may have given the learners false confidence in their speech abilities and led them to believe that they were replicating the sounds accurately, when, in reality, the precision of their production may not have matched their perception.

As has been discussed in the RQ1 section, a better implicit learning of sequence may have helped L2 learners effortlessly familiarize themselves with their own speech pattern. This familiarity may have fostered a sense of ease and confidence in their language abilities, as they become accustomed to the rhythms and sounds of their own speech. Consequently, this comfort could have caused them to overestimate their speaking abilities, mistaking familiarity for proficiency (cf. see Ortega et al. 2022 for a similar discussion). Therefore, the participants in the current study might have not recognized the difference between their accustomed speech patterns and the native or target language features/patterns, leading to a gap between their perceived and actual proficiency.

Conversely, spending more time learning English in regular classes could have instilled a sense of underconfidence, as learners are constantly exposed to the complexities of the language, and the emphasis is on areas that need improvement. In fact, previous studies of L2 speech self-evaluations revealed that L2 learners' perceived satisfaction with their performances strongly influenced how they evaluated themselves (e.g., Isbell and Lee 2022). As they were constantly reminded of the gaps in their knowledge and skills, this might have contributed to them having less confidence. Moreover, better grammar knowledge might have led to underestimation because the learners who were more aware of the

detailed grammatical rules may have become more self-conscious about making errors. In the current study, the participants had access to their own speech and paid close attention to what they said. Therefore, when striving for accuracy, their focus on any mistakes that they heard might have caused them to notice their weak points more overtly, thus causing them to judge their speaking skills more harshly.

Two cognitive IDs, audio-motor integration and phonological memory, were also associated with underconfidence. Learners with advanced audio–motor integration are more likely to be adept at synchronizing their motor processes with auditory inputs, which is a pivotal skill for language production. This synchronization not only assists in perceiving sounds, but also in replicating sound sequences and patterns accurately, which is essential in language learning and in musical training (Patel and Iversen 2014). Given that audio–motor integration underpins the coordination between auditory perception and the motor planning required for sound production, we might asume that learners with well-developed audio-motor integration abilities would be particularly skilled at emulating the speech patterns of a target language. However, this precise mimicry could increase their awareness of discrepancies between their own speech production and that of native speakers, potentially leading to an underestimation of their own language skills as they focus on even minor deviations from the ideal pronunciation (Flaugnacco et al. 2014; Gordon et al. 2015). L2 learners with stronger phonological memories have the advantage of retaining phonological information for more in-depth decoding. Therefore, the participants in this study may have been more able to critically evaluate their speech compared to the benchmarks due to their enhanced memory capacity. Research has shown that the integration of sound and motor execution is essential for success in L2 acquisition, and is beneficial for different types of linguistic training (Brekelmans et al. 2022; Li and DeKeyser 2017; Saito et al. 2021). However, this detailed understanding and increased awareness might led learners to set extremely high goals. If they compare their speech to that of native speakers and fall short, they might think that their skills do not meet the desired level of quality.

The age of learning and the amount of conversation per week had different influences depending on the dimensions. Specifically, these factors were associated with a tendency toward overconfidence in the self-assessments of comprehensibility (e.g., see Li 2018 for a similar result with age), yet these same factors were linked to underconfidence when evaluating accentedness. This suggests that earlier exposure to language learning and frequent conversational practice may boost learners' self-perceptions of their ability to be understood, possibly due to the cumulative effect of extended practice over time. However, these factors did not appear to translate into confidence regarding accent, as the learners may have considered the elusive nuances of native-like pronunciation to be a challenge. By contrast, working memory showed the opposite trend, as it was linked to underconfidence in comprehensibility but overconfidence in accentedness. This could imply that learners with stronger working memory capacities may have been more critical of their ability to convey meaning effectively, perhaps due to more extensive self-monitoring, and they may also have overestimated their pronunciation skills, potentially overlooking subtle phonological details that characterize native speakers' accents.

Overall, the variables linked to overestimation may foster a general sense of communicative efficacy, while those associated with underestimation may reflect a heightened awareness of linguistic precision and the gap between a learners' current abilities and the target language norms. Having said this, the variance explained by the variables in each model was small (4.1% and 3.1%, respectively), and not all the ID variables were found to be statistically significant in either the mixed-effect modeling or the regression analyses. Therefore, the influence of learner variables such as experiential and cognitive IDs might be small to medium at most (Ma and Winke 2019; Ross 1998; Suzuki 2015). Since studies in the field of psychology have indicated that the perceived difficulty of completing a task and the metacognitive awareness of the skill influence the level of confidence (e.g., Dunning et al. 1989; Burson et al. 2006), the main factor in the miscalibration may have been

psychological, such as the L2 learners' degree of satisfaction with their own pronunciation and/or metacognitive profile; for example, the value they placed on L2 speaking and pronunciation (Isbell and Lee 2022). In order to paint a fuller picture of the factors that affect the self-evaluation biases, further research with a wider range of IDs is required.

*6.4. Limitations and Future Resarch Direction*

Several methodological limitations need to be addressed. First, the participant population, drawn from a single demographic, may not capture the broader spectrum of L2 learners, which limits the generalizability of the findings (Suzuki 2015; Trofimovich et al. 2016). Second, the study's methodology, which emphasized certain cognitive IDs, may have neglected other influential psychological or sociocultural factors that have an impact on self-evaluations of L2 speech. Second, the results related to the role of phonological memory may be inconclusive because the language of the cognitive tests used to measure participants' phonological memory (adopted from Gathercole et al. 2001) was only in L2 (i.e., English). Existing research on the nature and impact of learners' phonological memory on L2 learning has suggested that the scores of phonological memory measured through the non-word stimuli created based on the L2 phonological system may not be an accurate representation of its language-independent component due to the influence of learners' L2 knowledge (e.g., phonological features and semantic aspects of L2 lexical items; see French and O'Brien 2008; Van Der Lely and Gallon 2006). Therefore, to capture learners' language-independent phonological memory, both L2-based non-word items and non-words generated from participants' unfamiliar language need to be prepared for the task. These two types of phonological memory scores (i.e., language-dependent phonological memory and language-independent phonological memory scores) may help us further understand the role of phonological memory in learners' self-evaluation behavior. In addition, the methodology allowed the participants to listen to their own speech for self-evaluation. However, other studies have often asked participants to self-evaluate without access to their speech (e.g., Saito et al. 2020b). This methodological choice may have influenced the self-assessment outcomes, which suggests that future studies should investigate the impact of the participants' access or lack of access to their own speech on the accuracy of self-evaluations (Isbell and Lee 2022). Expanding the research to include a more diverse participant base and comparing different self-assessment conditions could provide a more comprehensive understanding of the processes involved in self-evaluations of L2 speech.

## 7. Conclusions and Implication

In summary, the current study aimed to address the gap in research by investigating the influence of cognitive IDs on L2 speech self-evaluation accuracy and exploring their role in the mismatch between self-assessment and listener evaluation. The findings reveal weak correlations between participants' self-evaluation and native English listeners' evaluation of both comprehensibility and accentedness. This aligns with previous research indicating L2 learners' inability to accurately perceive their own speech. Notably, participants who were perceived as more comprehensible and less accented tended to underestimate their own speech, while those perceived less favorably tended to overestimate their abilities, confirming the presence of the Dunning–Kruger effect.

The study has also examined the impact of linguistic proficiency, experiential factors, and cognitive IDs on self-assessment accuracy. Linguistic proficiency, particularly in grammar and vocabulary, was found to contribute to better calibration, suggesting that learners with stronger language skills show greater accuracy in evaluating their own speech. On the other hand, cognitive IDs such as phonological memory and implicit learning played a role in the accurate self-evaluation of accentedness, but not comprehensibility. Learners with better implicit learning abilities tended to overestimate their accentedness, possibly due to their familiarity with their own speech patterns. However, there was no significant influence of cognitive IDs on comprehensibility, indicating that factors beyond pronunciation components contribute to this construct.

Exploratory analysis revealed associations between learner IDs and overconfidence or underconfidence in self-evaluation. Study abroad experience, vocabulary knowledge, and perceptual acuity were linked to overestimation, while English classes per week, grammar knowledge, and speech proficiency were associated with underestimation. These findings suggest that learners' perceptions of their language abilities are influenced by various cognitive and experiential factors, which may lead to overestimation or underestimation depending on individual characteristics. However, it is important to note that the variance explained by the ID variables was relatively small, indicating that other factors such as psychological factors and metacognitive awareness may also play a significant role in self-evaluation biases. Further research with a wider range of IDs is needed to fully understand the factors influencing L2 speech self-evaluation.

Nonetheless, these findings have important implications for language pedagogy. For instance, since existing studies prove that accurate self-assessment of comprehensibility can be improved through peer-assessment activities (Tsunemoto et al. 2022), and increased practice of speaking in the classroom and in extracurricular activities (Saito et al. 2020b), the current study suggests that language teachers might wish to raise awareness of the crucial role of lexicogrammatical knowledge in speech comprehensibility during such activities. Furthermore, in order to increase accuracy in the self-perception of both accentedness and comprehensibility, it is crucial for learners to understand the components of those constructs, given that the self-assessment literature posits that becoming familiar with the evaluation criteria helps learners assess their performance with more detail and specificity, resulting in alignment with others' assessments (e.g., Kissling and O'Donnell 2015). Such instruction may be particularly effective for those who are highly familiar with their own L2 speech patterns due to their better implicit learning ability, resulting in the overestimation of their accentedness. By raising awareness of the key phonological features for L2 learners, such learners may eventually be able to converge with others' evaluations.

**Funding:** This research was funded by Japan Grant-in-Aid for Scientific Research: Research Activity Start-up (grant number: 21K20019).

**Institutional Review Board Statement:** An ethical review and approval for this study were not required as Juntendo University, with which the researcher is affiliated, had previously secured ethical approval from the students upon their enrollment. This study engaged participants from that pre-approved pool and received the Ethics Committee's consent to collect linguistic and cognitive behavioral data from them.

**Informed Consent Statement:** Informed consent was obtained from all subjects involved in the study.

**Data Availability Statement:** The data presented in this study are available on request from the corresponding author.

**Acknowledgments:** I would like to extend my heartfelt gratitude to Takumi Uchihara, whose guidance and support were invaluable to the completion of this project. My sincere thanks also go to Satsuki Kurokawa and Kotaro Takizawa for their assistance and contributions that were crucial in the development of this research. Additionally, I am deeply grateful to Kazuya Saito for his expert advice and unwavering encouragement throughout this journey. Their collective knowledge, insight, and dedication have been a great source of inspiration and have significantly enriched this work.

**Conflicts of Interest:** The author declares no conflicts of interest.

## Appendix A

**Table A1.** Descriptive statistics.

|  | **M** | **SD** | **Minimum** | **Maximum** |
|---|---|---|---|---|
| Evaluation variables |  |  |  |  |
| Self-evaluation of comprehensibility | 4.47 | 1.95 | 1 | 9 |
| Self-evaluation of accentedness | 4.94 | 1.93 | 1 | 8 |

**Table A1.** *Cont.*

|  | **M** | **SD** | **Minimum** | **Maximum** |
|---|---|---|---|---|
| Comprehensibility | 5.66 | 2.07 | 1 | 9 |
| Accentedness | 4.42 | 1.96 | 1 | 9 |
| Confidence score (compressibility) | −1.19 | 2.58 | −8 | 5 |
| Confidence score (accentedness) | 0.515 | 2.45 | −7 | 6 |
| Distance score (compressibility) | 2.13 | 1.86 | 0 | 8 |
| Distance score (accentedness) | 1.80 | 1.63 | 0 | 7 |
| Individual differences variables |  |  |  |  |
| Age of Learning | 9.81 | 3.04 | 2 | 13 |
| English classes per week (h) | 1.89 | 2.62 | 0 | 15 |
| Study Abroad experience (yes/no) | 1.58 | 0.497 | NA | NA |
| Conversation per week (h) | 0.800 | 2.49 | 0 | 20 |
| Vocabulary knowledge (%) | 68.9 | 7.81 | 50 | 100 |
| Grammar knowledge (%) | 71.1 | 10.3 | 46 | 100 |
| Speech proficiency | 4.47 | 1.95 | 1 | 9 |
| Perceptual acuity | 36.4 | 16.4 | 3.6 | 97.4 |
| Audio–motor integration | 41.5 | 5.45 | 29.1 | 50.5 |
| Working memory | 14.0 | 2.12 | 6 | 16 |
| Phonological memory | 13.8 | 2.77 | 6 | 19 |
| Implicit learning | 93.1 | 69.9 | −117 | 271 |

**Table A2.** Correlation results of distance score (comprehensibility).

| Variable |  | 1 | 2 | 3 | 4 | 5 | 6 | 7 | 8 | 9 | 10 | 11 | 12 |
|---|---|---|---|---|---|---|---|---|---|---|---|---|---|
| 1. Distance score (comprehensibility) |  |  |  |  |  |  |  |  |  |  |  |  |  |
| 2. Age of learning | r | −0.079 |  |  |  |  |  |  |  |  |  |  |  |
|  | p | 0.442 |  |  |  |  |  |  |  |  |  |  |  |
| 3. English classes per week | r | −0.077 | −0.048 |  |  |  |  |  |  |  |  |  |  |
|  | p | 0.452 | 0.644 |  |  |  |  |  |  |  |  |  |  |
| 4. Study abroad experience | r | 0.17 | 0.139 | −0.11 |  |  |  |  |  |  |  |  |  |
|  | p | 0.096 | 0.175 | 0.282 |  |  |  |  |  |  |  |  |  |
| 5. Conversation per week | r | 0.052 | −0.12 | 0.204 | −0.205 |  |  |  |  |  |  |  |  |
|  | p | 0.611 | 0.242 | 0.045 | 0.044 |  |  |  |  |  |  |  |  |
| 6. Grammar knowledge | r | 0.075 | −0.098 | −0.15 | −0.144 | −0.013 |  |  |  |  |  |  |  |
|  | p | 0.468 | 0.34 | 0.141 | 0.16 | 0.897 |  |  |  |  |  |  |  |
| 7. Vocabulary knowledge | r | −0.175 | −0.142 | −0.033 | −0.04 | 0.118 | 0.081 |  |  |  |  |  |  |
|  | p | 0.087 | 0.164 | 0.745 | 0.695 | 0.249 | 0.431 |  |  |  |  |  |  |
| 8. Speech proficiency | r | 0.352 | −0.063 | 0.031 | 0.113 | 0.061 | 0.037 | 0.081 |  |  |  |  |  |
|  | p | <0.001 | 0.543 | 0.761 | 0.269 | 0.552 | 0.721 | 0.433 |  |  |  |  |  |
| 9. Perceptual acuity | r | 0.263 | −0.037 | 0.131 | 0.167 | −0.156 | 0.02 | −0.026 | 0.203 |  |  |  |  |
|  | p | 0.009 | 0.719 | 0.202 | 0.101 | 0.128 | 0.844 | 0.803 | 0.046 |  |  |  |  |
| 10. Audio-motor integration | r | −0.016 | 0.032 | 0.029 | −0.179 | 0.058 | −28.7 | −0.067 | −0.134 | −0.284 |  |  |  |
|  | p | 0.877 | 0.753 | 0.781 | 0.08 | 0.574 | 0.998 | 0.515 | 0.19 | 0.005 |  |  |  |
| 11. Working memory | r | 0.053 | −0.034 | 0.147 | −0.085 | 0.04 | 0.135 | −0.053 | 0.001 | −0.102 | 0.179 |  |  |
|  | p | 0.608 | 0.743 | 0.152 | 0.408 | 0.7 | 0.186 | 0.603 | 0.989 | 0.32 | 0.079 |  |  |
| 12. Phonological memory | r | −0.097 | −0.09 | 0.09 | −0.081 | −0.033 | −0.006 | 0.092 | −0.003 | −0.162 | 0.349 | 0.156 |  |
|  | p | 0.344 | 0.379 | 0.381 | 0.432 | 0.75 | 0.952 | 0.372 | 0.978 | 0.113 | <0.001 | 0.127 |  |
| 13. Implicit learning | r | −0.114 | −0.093 | −0.159 | −0.101 | 0.011 | 0.165 | 0.186 | −0.069 | 0.1 | 0.03 | 0.107 | 0.125 |
|  | p | 0.267 | 0.366 | 0.12 | 0.323 | 0.917 | 0.106 | 0.068 | 0.499 | 0.332 | 0.768 | 0.298 | 0.221 |

**Table A3.** Correlation results of distance score (accentedness).

| Variable |  | 1 | 2 | 3 | 4 | 5 | 6 | 7 | 8 | 9 | 10 | 11 | 12 |
|---|---|---|---|---|---|---|---|---|---|---|---|---|---|
| 1. Distance score (comprehensibility) |  |  |  |  |  |  |  |  |  |  |  |  |  |
| 2. Age of learning | r | 0.064 |  |  |  |  |  |  |  |  |  |  |  |
|  | p | 0.534 |  |  |  |  |  |  |  |  |  |  |  |
| 3. English classes per week | r | −0.22 | −0.048 |  |  |  |  |  |  |  |  |  |  |
|  | p | 0.031 | 0.644 |  |  |  |  |  |  |  |  |  |  |
| 4. Study abroad experience | r | 0.097 | 0.139 | −0.11 |  |  |  |  |  |  |  |  |  |
|  | p | 0.345 | 0.175 | 0.282 |  |  |  |  |  |  |  |  |  |
| 5. Conversation per week | r | −0.036 | −0.12 | 0.204 | −0.205 |  |  |  |  |  |  |  |  |
|  | p | 0.725 | 0.242 | 0.045 | 0.044 |  |  |  |  |  |  |  |  |
| 6. Grammar knowledge | r | 0.042 | −0.098 | −0.15 | −0.144 | −0.013 |  |  |  |  |  |  |  |
|  | p | 0.684 | 0.34 | 0.141 | 0.16 | 0.897 |  |  |  |  |  |  |  |
| 7. Vocabulary knowledge | r | −0.089 | −0.142 | −0.033 | −0.04 | 0.118 | 0.081 |  |  |  |  |  |  |
|  | p | 0.386 | 0.164 | 0.745 | 0.695 | 0.249 | 0.431 |  |  |  |  |  |  |

**Table A3.** *Cont.*

| Variable | | 1 | 2 | 3 | 4 | 5 | 6 | 7 | 8 | 9 | 10 | 11 | 12 |
|---|---|---|---|---|---|---|---|---|---|---|---|---|---|
| 8. Speech proficiency | *r* | 0.029 | −0.063 | 0.031 | 0.113 | 0.061 | 0.037 | 0.081 | | | | | |
| | *p* | 0.776 | 0.543 | 0.761 | 0.269 | 0.552 | 0.721 | 0.433 | | | | | |
| 9. Perceptual acuity | *r* | 0.283 | 0.092 | 0.149 | 0.148 | −0.066 | −0.054 | −0.081 | 0.074 | | | | |
| | *p* | 0.005 | 0.369 | 0.146 | 0.147 | 0.521 | 0.598 | 0.431 | 0.472 | | | | |
| 10. Audio-motor integration | *r* | −0.239 | 0.032 | 0.029 | −0.179 | 0.058 | −280.7 | −0.067 | −0.134 | −0.392 | | | |
| | *p* | 0.018 | 0.753 | 0.781 | 0.08 | 0.574 | 0.998 | 0.515 | 0.19 | <0.001 | | | |
| 11.Working memory | *r* | 0.11 | −0.034 | 0.147 | −0.085 | 0.04 | 0.135 | −0.053 | 0.001 | 0.019 | 0.179 | | |
| | *p* | 0.282 | 0.743 | 0.152 | 0.408 | 0.7 | 0.186 | 0.603 | 0.989 | 0.856 | 0.079 | | |
| 12. Phonological memory | *r* | −0.175 | −0.09 | 0.09 | −0.081 | −0.033 | −0.006 | 0.092 | −0.003 | −0.201 | 0.349 | 0.156 | |
| | *p* | 0.086 | 0.379 | 0.381 | 0.432 | 0.75 | 0.952 | 0.372 | 0.978 | 0.049 | <0.001 | 0.127 | |
| 13. Implicit learning | *r* | 0.185 | −0.093 | −0.159 | −0.101 | 0.011 | 0.165 | 0.186 | −0.069 | 0.04 | 0.03 | 0.107 | 0.125 |
| | *p* | 0.07 | 0.366 | 0.12 | 0.323 | 0.917 | 0.106 | 0.068 | 0.499 | 0.698 | 0.768 | 0.298 | 0.221 |

**Table A4.** Correlation results of confidence score (comprehensibility).

| Variable | | 1 | 2 | 3 | 4 | 5 | 6 | 7 | 8 | 9 | 10 | 11 | 12 |
|---|---|---|---|---|---|---|---|---|---|---|---|---|---|
| 1. Distance score (comprehensibility) | | | | | | | | | | | | | |
| 2. Age of learning | *r* | −0.025 | | | | | | | | | | | |
| | *p* | 0.811 | | | | | | | | | | | |
| 3. English classes per week | *r* | −0.089 | −0.048 | | | | | | | | | | |
| | *p* | 0.385 | 0.644 | | | | | | | | | | |
| 4. Study abroad experience | *r* | −0.183 | 0.139 | −0.11 | | | | | | | | | |
| | *p* | 0.073 | 0.175 | 0.282 | | | | | | | | | |
| 5. Conversation per week | *r* | 0.081 | −0.12 | 0.204 | −0.205 | | | | | | | | |
| | *p* | 0.429 | 0.242 | 0.045 | 0.044 | | | | | | | | |
| 6. Grammar knowledge | *r* | −0.058 | −0.098 | −0.15 | −0.144 | −0.013 | | | | | | | |
| | *p* | 0.571 | 0.34 | 0.141 | 0.16 | 0.897 | | | | | | | |
| 7. Vocabulary knowledge | *r* | 0.14 | −0.142 | −0.033 | −0.04 | 0.118 | 0.081 | | | | | | |
| | *p* | 0.171 | 0.164 | 0.745 | 0.695 | 0.249 | 0.431 | | | | | | |
| 8. Speech proficiency | *r* | −0.684 | −0.063 | 0.031 | 0.113 | 0.061 | 0.037 | 0.081 | | | | | |
| | *p* | <0.001 | 0.543 | 0.761 | 0.269 | 0.552 | 0.721 | 0.433 | | | | | |
| 9. Perceptual acuity | *r* | −0.22 | −0.037 | 0.131 | 0.167 | −0.156 | 0.02 | −0.026 | 0.203 | | | | |
| | *p* | 0.03 | 0.719 | 0.202 | 0.101 | 0.128 | 0.844 | 0.803 | 0.046 | | | | |
| 10. Audio-motor integration | *r* | 0.089 | 0.032 | 0.029 | −0.179 | 0.058 | −28.7 | −0.067 | −0.134 | −0.284 | | | |
| | *p* | 0.387 | 0.753 | 0.781 | 0.08 | 0.574 | 0.998 | 0.515 | 0.19 | 0.005 | | | |
| 11.Working memory | *r* | −0.039 | −0.034 | 0.147 | −0.085 | 0.04 | 0.135 | −0.053 | 0.001 | −0.102 | 0.179 | | |
| | *p* | 0.702 | 0.743 | 0.152 | 0.408 | 0.7 | 0.186 | 0.603 | 0.989 | 0.32 | 0.079 | | |
| 12. Phonological memory | *r* | 0.02 | −0.09 | 0.09 | −0.081 | −0.033 | −0.006 | 0.092 | −0.003 | −0.162 | 0.349 | 0.156 | |
| | *p* | 0.844 | 0.379 | 0.381 | 0.432 | 0.75 | 0.952 | 0.372 | 0.978 | 0.113 | <0.001 | 0.127 | |
| 13. Implicit learning | *r* | 0.221 | −0.093 | −0.159 | −0.101 | 0.011 | 0.165 | 0.186 | −0.069 | 0.1 | 0.03 | 0.107 | 0.125 |
| | *p* | 0.029 | 0.366 | 0.12 | 0.323 | 0.917 | 0.106 | 0.068 | 0.499 | 0.332 | 0.768 | 0.298 | 0.221 |

**Table A5.** Correlation results of confidence score (accentedness).

| Variable | | 1 | 2 | 3 | 4 | 5 | 6 | 7 | 8 | 9 | 10 | 11 | 12 |
|---|---|---|---|---|---|---|---|---|---|---|---|---|---|
| 1. Distance score (comprehensibility) | | | | | | | | | | | | | |
| 2. Age of learning | *r* | −0.036 | | | | | | | | | | | |
| | *p* | 0.724 | | | | | | | | | | | |
| 3. English classes per week | *r* | −0.181 | −0.048 | | | | | | | | | | |
| | *p* | 0.076 | 0.644 | | | | | | | | | | |
| 4. Study abroad experience | *r* | −0.137 | 0.139 | −0.11 | | | | | | | | | |
| | *p* | 0.181 | 0.175 | 0.282 | | | | | | | | | |
| 5. Conversation per week | *r* | 0.056 | −0.12 | 0.204 | −0.205 | | | | | | | | |
| | *p* | 0.589 | 0.242 | 0.045 | 0.044 | | | | | | | | |
| 6. Grammar knowledge | *r* | 0.052 | −0.098 | −0.15 | −0.144 | −0.013 | | | | | | | |
| | *p* | 0.613 | 0.34 | 0.141 | 0.16 | 0.897 | | | | | | | |
| 7. Vocabulary knowledge | *r* | 0.064 | −0.142 | −0.033 | −0.04 | 0.118 | 0.081 | | | | | | |
| | *p* | 0.534 | 0.164 | 0.745 | 0.695 | 0.249 | 0.431 | | | | | | |
| 8. Speech proficiency | *r* | −0.094 | −0.063 | 0.031 | 0.113 | 0.061 | 0.037 | 0.081 | | | | | |
| | *p* | 0.359 | 0.543 | 0.761 | 0.269 | 0.552 | 0.721 | 0.433 | | | | | |
| 9. Perceptual acuity | *r* | 0.013 | 0.092 | 0.149 | 0.148 | −0.066 | −0.054 | −0.081 | 0.074 | | | | |
| | *p* | 0.899 | 0.369 | 0.146 | 0.147 | 0.521 | 0.598 | 0.431 | 0.472 | | | | |
| 10. Audio-motor integration | *r* | −0.099 | 0.032 | 0.029 | −0.179 | 0.058 | −280.7 | −0.067 | −0.134 | −0.392 | | | |
| | *p* | 0.333 | 0.753 | 0.781 | 0.08 | 0.574 | 0.998 | 0.515 | 0.19 | <0.001 | | | |
| 11.Working memory | *r* | 0.012 | −0.034 | 0.147 | −0.085 | 0.04 | 0.135 | −0.053 | 0.001 | 0.019 | 0.179 | | |
| | *p* | 0.905 | 0.743 | 0.152 | 0.408 | 0.7 | 0.186 | 0.603 | 0.989 | 0.856 | 0.079 | | |
| 12. Phonological memory | *r* | −0.136 | −0.09 | 0.09 | −0.081 | −0.033 | −0.006 | 0.092 | −0.003 | −0.201 | 0.349 | 0.156 | |
| | *p* | 0.185 | 0.379 | 0.381 | 0.432 | 0.75 | 0.952 | 0.372 | 0.978 | 0.049 | <0.001 | 0.127 | |
| 13. Implicit learning | *r* | 0.091 | −0.093 | −0.159 | −0.101 | 0.011 | 0.165 | 0.186 | −0.069 | 0.04 | 0.03 | 0.107 | 0.125 |
| | *p* | 0.373 | 0.366 | 0.12 | 0.323 | 0.917 | 0.106 | 0.068 | 0.499 | 0.698 | 0.768 | 0.298 | 0.221 |

## Notes

1    Isbell and Lee (2022), whose study is a conceptual replication of Trofimovich et al. (2016), speculated that the discrepancy in the correlation results between their study and that of Trofimovich et al. (2016) was to do with the differences in rater experience—the former were novice and inexperienced raters of L2 Korean speech, whereas the latter were expert raters with extensive experience of teaching and assessing L2 English.

2    The following are details of the variety in the participants' proficiency: A2 ($n = 1$; 1%), B1 ($n = 28$; 29%), B2 ($n = 44$; 45%), C1 ($n = 24$; 25%).

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
