# Peer review of "Delving into L2 Learners’ Perspective: Exploring the Role of Individual Differences in Self-Evaluation of L2 Speech Learning"

_languages, doi:10.3390/languages9030109_

Round 1

Reviewer 1 Report

Comments and Suggestions for Authors

This is a well-conducted study on self-assessment of L2 speech, which explores the role of IDs. It is overall clear and well-structured. However, the language and style of the manuscript should be improved. You’ll find my comments in what follows:

Abstract

Line 8: the acronym L2 should appear in line 7 (second language self-perception).

Line 9: I think it should read ‘the current study investigated the roles of individual differences (IDs; especially experiential and 9 cognitive IDs) influence the ON learners' self-assessment accuracy.

Lines 10: I would say L2 speech samples.

Line 11. It should be rewritten.

Line 13: I would omit the word amount, and just say ‘the gap between…’

Introduction

The text needs to be revised for accuracy. This is just an example, but the whole article needs revision: Lines 26-29: “Despite its value for acquisition, numerous studies have reported these estimations often 26 miss the mark (Foote 2010; Suzuki 2015) while some indicates learners can accurately 27 estimate their L2 proficiency (Brantmeier and Vanderplank, 2008; Luoma, 2012; Préfon- 28 taine, 2013): I would move the part that some studies indicate that learners can self-estimate their proficiency at the beginning. A comma should also be used to separate the two clauses. There is also a problem with subject-verb agreement (some indicates).

Lines 29-30: The sentence is not grammatical. A main verb is needed.

Lines 44-45: the sentence is not grammatical.

Line 129: spelling mistake: calibration.

Line 130: spelling mistake: miscalibration.

These are just examples. The whole article should be improved as regards language.

Line 39: Do you mean IDs here? I’d be clearer here.

Background

Line 130: Avoid asking questions.

Line 151: I don’t think it is necessary to remind the reader that this is the focus of the current study. I think that it is clear already.

Line 180: is it better? Or the opposite?

The current study

For me this should be section 3, and not 2.4.

I think that this section includes information that should be part of the literature review.

Line 231: I would be more explicit in the formulation of the RQ. I would include the specific cognitive abilities tested in the formulation of the RQ.

I would include the two RQs together and then justify the decisions taken.

Materials and Method

Lines 271-273: ‘At the time of the project, the participants 271 equally received English education in junior high and high school, and enrolled in 272 various undergraduate courses in at universities in Japan.’ What does this represent as regards of number of hours of exposure to English?

Line 273: You need to be more specific about the learners’ proficiency level.

Lines 280-284: Is this the order in which participants did the tests? If so, say it. If it is not the case, I think this information should be included in the methods. Did all participants perform the tests in the same order?

I would maybe have a section called instruments or tests or materials, this could be 3.3. And inside it you could have subsection for the different tests. The first one would be the grammaticality judgement task.

Line 399: Several studies have asked learners to listen to their own recordings prior to the self-assessment. I would check this and rewrite accordingly.

Line 404: Did you explain participants what comprehensibility and accentedness mean? Can you give more details on the instructions given to participants?

For me 3.6 and 3.7 should be a single section.

Lines 462-464: Shouldn’t it be moved to the results?

Results

Section 4.1: You need to explain what these correlations suggest.

Lines 495-496: Both correlations are very weak.

Line 501: This correlation is also quite weak.

Line 505-508: This should be moved to the ‘Analysis’ section.

Line 515: Why was group included as a random factor and not as a fixed effect?

Discussion

In the discussion, I would start reminding the reader of the research questions, then what you found and then compare it with previous results. There is no need to start the discussion section in the same way that you would start an introduction.

Some of the information included in the discussion part should actually be placed in the results section.

The results should be further discussed, especially those for RQ1.

I would have liked to see a section with the conclusion. In such a section you could have explained the pedagogical implications of your findings.

References

References should be revised. For example, the following reference is repeated: Li Mushi. 2018. Know thyself? Self- vs. other-assessment of second language pronunciation. (Doctoral dissertation, Boston University). 1049 Available from ProQuest Dissertations and Theses Global database.

Comments on the Quality of English Language

Language and style should be improved

Reviewer 2 Report

Comments and Suggestions for Authors

General comments:

• The style and register are appropriate, and notwithstanding the errors, the language is generally clear and comprehensible: sentences are generally short (sometimes even too short, though there is at least one notable exception), and the language is neutral, often formulaic even. However, the quality of the language needs addressing, as I mention in the following section. 

• The study presented asks some interesting questions and provides some interesting answers – it is refreshing to see individual learner differences considered in studies involving larger numbers (n=97) of students.

• The substance of the article is solid – the lit review, the methodology, the analysis of the results and so forth. The literature review is extensive, with over 90 references.

• I’m not sure the template is always followed – for example the .dot provided by Languages uses fully justified text, this manuscript is left justified only. And some of the line spacing between sections is out.

• The structure of the article – more specifically the numbering and titles of the sections – is a bit misleading (see next part).

Specific comments:

(most of my specific comments are made in my annotated version of the manuscript – I think that’s easier for all concerned)

• The lit review spreads over the introduction (section 1) and “Background” (section 2), and the description of the study starts in the middle of section 2. I would suggest that a more logical way to approach the presentation would be:

o A shorter, more general introduction, establishing the need for this research (which I do not call into question – it is interesting and useful). This introduction would of course contain some references, especially the more classic ones on intelligibility, assessment, speaking, learner differences and learner cognition, etc.

o The next section – still called “Background” would be exclusively devoted to a literature review, with subsections labelled basically as they are now, developing the themes mentioned in the intro., and one of the su-sections could be on learner-assessment, which is the main content of the intro. As it currently stands.

o I would suggest that a better title for 2.4 would be “Rationale for the current study” or sth similar, perhaps even mention the RQs in the heading, or have a separate heading mentioning RQs? –That is what 2.4 actually is - rationale & RQs. I can see the logic, as it does fall within section 2 “Background”, but it doesn’t help readers navigate the article. Perhaps group the RQs, maybe in a sub-section, or at least leave a line, or use bold. Both research questions concern IDs, is it really necessary to separate them?

• I have a problem with the variables mentioned as being “the acoustic dimension” (L368): “formant, duration, risetime, and pitch” are not the acoustic correlates of the speech signal – those are formant structure / spectral form (you need a spectrogram for that), duration (ms), amplitude / intensity (dB) and F0 (which is perceived as pitch).

• Some of the ideas put forward in the discussion section are a little over-stated. I would prefer to see more toned-down language, using modal verbs and expressions such as “this could be due to”, etc. I have only highlighted one example in the section, but it is a cumulative effect.

Comments on the Quality of English Language

·         More conscientious proofing of the article of a whole before re-submission is a must

·         The style and register are appropriate, and notwithstanding the errors, the language is generally clear and comprehensible: sentences are generally short (sometimes even too short, though there is at least one notable exception), and the language is neutral, often formulaic even. However, the quality of the language and the number of punctuation errors, grammatical errors, etc. is unacceptable. I have done my best to find and correct these errors, but it shouldn’t really be the job of a reviewer. Before a second draft is submitted, it must be:

o   Re-read and corrected for grammar mistakes

o   Thoroughly proofed for typos, especially punctuation errors in the references both in-text and in the bibliography

This is why I have included a copy of the PDF of the manuscript with highlighted text and comments. Nothing major, but there are a lot of typos. 

Reviewer 3 Report

Comments and Suggestions for Authors

A very interesting paper that is very well located within extensive current literature and research trends.

SOME sentences from the results would work really well in the abstract. They are clear and direct:

THE STUDY explored the role of IDs in the 
miscalibration of L2 speech evaluation AND took an exploratory L747
approach to examine which learner IDs contribute to overconfidence and
underconfidence in self-evaluation

You might like to include the  correlation and multiple regression analyses in the abstract.

So much detailed work has  gone into this study . As an applied linguist I am looking for the pedagogical implications. Now we know what you have found what do you recommend to students what would you say to teachers. This is an essential part of the puzzle that needs to be addressed so the work can be used to further L2 teaching and learning not just the research side.

 ' to examine the discrepancies between self- and other L491
evaluations of L2 speech comprehensibility and accentedness, correlation
analyses were conducted'

to identify the role of cognitive IDs in overconfident and underconfident L 565
estimation of own speech quality, a series of multiple regression analyses were
performed

Looking forward to seeing it in print.

Comments on the Quality of English Language

I have remarked in line order and made some suggestions  IN CAPITALS and used (  ) to mark deletions required. /\ was used when words were added to the original. In most cases the capitals are to be in lower case in edited version unless convention requires otherwise

L9 HOW the roles of individual differences (IDs; especially experiential and

L10 L2 speech elicited from 97 Japanese learners English WAS

L26 reported THAT these estimation

L27 while OTHERS indicate(s) learners can accurately

L30  SUGGESTS that approximating

that can impact ( the )  self-assessment L89

L120  from (the )wrong self-evaluation

L131 that their ( belief )ATTITUDE towards

, exiSting L138

examined 63 Japanese learners self-evaluation of Chinese (’s )(on) reading skillS. L155

Pertaining to HOW IDs impact self-evaluation in L2 pronunciation and speech, similar L160

 in ( the0 L2 self-evaluation has been shifting towards L177

 satisfaction with pronunciation skillS emerged as a L188

AN ID paradigm L191

 identified A L 208

 i.e., working memory, phonological memory, implicit sequence L234

, (They were) IT WAS included 244

thought to BE a basis for learning and 246

to be a L258 AN ESSENTIAL vial? ability   Sorry   there may be a typo and I cannot understand ... vial is a small bottle in pharmacy? if there is another meaning it may be too obscure or outdated?

 L271 (equally) received EQUAL AMOUNTS OF English education in junior high and high school,

 hearing ( and) OR reading L276

lexicogrammatical( ly) accuracy, and the linguistic knowledge is assumed to be L291

, the participants/\' grammatical and lexical 292

IN ADDITION (the) linguistic knowledge is assumed to be L291

),/\ A timed grammaticality judgement task L294

 /\THE Partici-L 339

 (is) WAS to listen to the pairs, and then decide(d) whether the paired sequences were L340

 prompt- L346 ing participantS

 participantS famil- L348

 conducted (via) online. Prior to the main session, the raters L426

calculated: /\A confidence score and /\A distance score./\ THE Confidence score indicates the nature of L447

/\ THE confidence score was calculated by subtracting the mean raters’ 449

Frequently, L2 learners  incorrectly perceive their linguistic ability (e.g., Gaffney L651

there's please expand: the contraction is not conventional academic English

/\ IS a manifestation of the Dunning-Kruger effect when comparing self-evaluations to 654

proficiency /\THAT determines how accurately they can self-evaluate their comprehensibility. 730

o (the) overconfidence and 748

could have L784 instillED 

studIES identified that L2 learners’ perceived L787

trongLY influence on how they evaluate themselves L788

, they might think their skills do(es) not L818
